# EX-NVS: EXtreme Novel View Synthesis via Depth Watertight Mesh

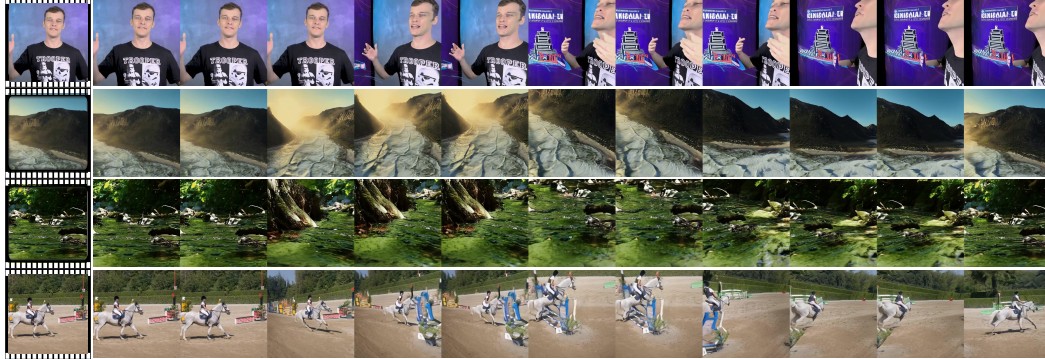

Input: Monocular Video  Output: Extreme Viewpoint Video

Figure 1: Our EX-NVS framework takes a monocular video as input and generates high-quality videos under extreme viewpoints. By leveraging the proposed Depth Watertight Mesh representation, it effectively handles occlusions in boundaries and ensures geometric consistency, enabling visually coherent and realistic results.

## Abstract

We introduce EX-NVS, a framework that addresses these challenges via a Depth Watertight Mesh (DW-Mesh) representation that explicitly models both visible and occluded regions, providing a robust geometric prior across viewpoints. Unlike traditional surface reconstruction methods that struggle with sparse visibility, our DW-Mesh ensures complete geometric coverage and maintains watertight properties essential for extreme viewpoint synthesis. To overcome the requirement for multi-view paired training data, we propose a simulated masking strategy that produces effective supervision from common monocular videos. A lightweight LoRA-based video diffusion adapter with novel linear aggregation capabilities integrates the DW-Mesh priors to synthesize high-quality, physically consistent, and temporally coherent videos. Extensive experiments demonstrate that EX-NVS outperforms state-of-the-art methods across a variety of metrics, with particularly strong improvements for extreme camera angles ranging from -90° to 90°, enabling practical extreme novel view synthesis.

## 1 Introduction

Recent advances in video generative models (Blattmann et al., 2023; Yang et al., 2024; et al., 2024; 2025a;b) enable high-quality, controllable video synthesis from text, images, and videos. Within this rapidly evolving field, camera-controllable video generation (He et al., 2024; Zhao et al., 2025; Liu et al., 2024) has emerged as a critical direction: enabling viewers to experience static or dynamic scenes from multiple viewpoints by simultaneously modeling spatial, temporal, and viewpoint dimensions. This capability underpins next-generation mixed reality experiences, free-viewpoint video systems, and immersive 3D content production that are becoming increasingly important in entertainment, education, and virtual collaboration.

However, generating camera-controllable videos for dynamic scenes with extreme viewpoints (e.g., $-90°$ to $90°$) remains one of the most challenging problems in computer vision. Current approaches fall into two main paradigms, each with fundamental limitations: 1) *Camera-based guidance* (He et al., 2024; Bai et al., 2024; Bahmani et al., 2025; Bai et al., 2025; Liu et al., 2023) uses camera parameters as implicit conditions, encoding position and orientation through ray maps (Hodge & Pedoe, 1994), positional embeddings (Mildenhall et al., 2020), or relative pose prompts. While these can generate videos with varying camera poses, they lack physical-consistent controllability and require extensive multi-view datasets with accurate camera calibration, limiting their practical applicability. 2) *Geometry-based guidance* (Gu et al., 2025; YU et al., 2025; Xiao et al., 2025) leverages explicit 3D representations to enable viewpoint control. These approaches reconstruct 3D geometry (e.g., point clouds, meshes) from input frames using techniques like MVSNet (Yao et al., 2018) or recent point-map methods (Wang et al., 2024b; Zhang et al., 2024; Wang et al., 2025b;a), then render these representations from target cameras to guide generation. While they reduce dependence on camera-calibrated multi-view training data (YU et al., 2025), they face a critical limitation: incomplete representation of occluded regions. This leads to boundary artifacts under extreme viewpoints, compromising visual quality and physical consistency.

To address these limitations, we propose **EX-NVS**, a framework for transforming monocular videos into **EX**treme-viewpoint **N**ovel **V**iew **S**ynthesis (Fig. 1). Our approach represents a paradigm shift that bridges camera- and geometry-based methods, synthesizing convincing extreme-view videos from monocular input without multi-view training data, while achieving physically consistent viewpoint control with seamless boundary-occlusion continuity and robust temporal appearance coherence.

The key module of our framework is the *Depth Watertight Mesh* (DW-Mesh) representation, which serves as a comprehensive geometric prior to guide the video generation process. Unlike traditional surface reconstruction methods that struggle with sparse visibility across viewpoints, the DW-Mesh explicitly models both visible surfaces and occluded boundaries through its watertight structure, ensuring geometric consistency even under the most extreme camera movements. This representation provides reliable and complete visibility masks for every viewpoint, effectively handling occlusion transitions through its mathematically sound watertight formulation.

To overcome the fundamental challenge of multi-view training data scarcity, we introduce a novel *simulated masking strategy* that creates highly effective training samples from readily available monocular videos. This approach employs two synergistic techniques: (1) *Rendering Mask Generation*, which creates physically grounded visibility masks from our DW-Mesh to simulate novel viewpoint occlusions; and (2) *Tracking Mask Generation*, which ensures temporal consistency by tracking feature correspondences across frames. This strategy eliminates the need for expensive multi-view data collection while effectively simulating the full spectrum of extreme viewpoint challenges.

Finally, guided by the DW-Mesh priors, a lightweight *LoRA-based video diffusion adapter* with linear aggregation capabilities synthesizes high-quality videos with enhanced temporal coherence. This adapter efficiently integrates geometric information from the DW-Mesh with pre-trained video diffusion models through an innovative multi-input fusion mechanism, producing visually coherent and physically realistic results while maintaining computational efficiency. Our comprehensive experiments demonstrate that EX-NVS consistently outperforms state-of-the-art methods across different metrics and viewpoint ranges, with the performance gap widening significantly as camera angles become more extreme. Quantitative evaluations show substantial improvements in visual quality (FID), temporal coherence (FVD), and 3D consistency (PSNR), while user studies confirm superior perceptual quality and physical consistency. Importantly, our method produces more realistic and physically consistent videos, particularly for challenging viewpoints ranging from $-90°$ to $90°$, representing a significant advance in practical extreme viewpoint video synthesis capabilities.

In summary, our main contributions are:

1. We introduce the Depth Watertight Mesh (DW-Mesh) representation that explicitly models both visible and hidden regions with watertight properties, maintaining geometric consistency for extreme viewpoints where traditional surface reconstruction fails.

2. We develop a comprehensive simulated masking strategy combining rendering masks and tracking masks that enables effective training without requiring expensive multi-view video datasets, democratizing extreme viewpoint synthesis.

3. Extensive experiments demonstrate EX-NVS consistently outperforms existing methods across all metrics, with particularly strong improvements for extreme camera angles, enabling practical applications in immersive content creation.

## 2 RELATED WORK

**Scene Reconstruction.** Recent advances include neural representations like NeRF (Mildenhall et al., 2020), efficient methods such as 3D Gaussian Splatting (Kerbl et al., 2023), and dynamic scene modeling with Shape-of-Motion (Wang et al., 2024a). Approaches like DUSt3R (Wang et al., 2024b), X-Ray (Hu et al., 2024), CUT3R (Wang et al., 2025b), and VGGT (Wang et al., 2025a) have improved efficiency by reconstructing from uncalibrated images. However, these methods often struggle with occlusions and dynamic scenes. Our DW-Mesh explicitly models occluded regions to ensure geometric consistency during extreme viewpoint synthesis.

**Video Diffusion Models.** The field has evolved from early approaches like Make-A-Video (Singer et al., 2023) and Gen-1 (RunwayML, 2023) to more sophisticated models. SVD (Blattmann et al., 2023) and VideoCrafter (Chen et al., 2023; 2024) enhanced temporal coherence, while large-scale models such as Hunyuan Video (et al., 2024), CogVideoX (Yang et al., 2024), and Wan 2.1 (et al., 2025b) achieve impressive spatiotemporal consistency. Our framework builds on these capabilities to enable extreme-angle video generation with robust geometric and temporal coherence.

**Camera and Motion Control.** Various approaches enable camera movement in video synthesis, but face significant limitations under extreme viewpoints. CameraCtrl (He et al., 2024) uses camera parameter encoding but struggles with extreme viewpoints due to lack of geometric understanding. GCD (Van Hoorick et al., 2024) employs pose embeddings but requires domain-specific training. TrajectoryCrafter (YU et al., 2025) enables camera redirection using point clouds but suffers from incomplete geometry reconstruction that leads to artifacts under extreme viewing angles. ReCam-Master (Bai et al., 2025) extends T2V models with camera control but needs extensive multi-camera training data. Other approaches like MotionCtrl (Wang et al., 2024c), AnimateDiff (Guo et al., 2024; 2023), and DragNUWA (Yin et al., 2023) support basic camera effects without proper 3D geometric understanding. Our approach fundamentally addresses these limitations through comprehensive DW-Mesh representation that maintains watertight geometric properties, enabling high-quality novel view synthesis from monocular videos without multi-view training data while ensuring physical consistency under extreme camera movements.

## 3 OUR APPROACH

The goal of our EX-NVS framework is to generate a novel-view video $\hat{V} = \{\hat{I}_t\}_{t=1}^T$ from an input monocular video $V = \{I_s\}_{s=1}^S$ and a target camera trajectory $\{P_t\}_{t=1}^T$. It consists of three key steps: (1) constructing a DW-Mesh as a geometric prior to handle occlusions in boundaries, (2) generating training masks to simulate novel-view occlusions using monocular videos, and (3) using a lightweight video diffusion adapter to produce physically consistent and temporally coherent videos.

### 3.1 DEPTH WATERTIGHT MESH

Existing 3D representations for novel view synthesis typically focus on visible surfaces while neglecting occluded regions, leading to artifacts when rendering from extreme viewpoints. Our DW-Mesh addresses this unexplored limitation by implementing a geometric structure that maintains both visible and hidden surfaces through a watertight formulation. This technical design choice enables unified handling of scene topology across arbitrary camera positions without requiring explicit multi-view supervision.

### 3.1.1 DW-MESH CONSTRUCTION

As shown in Fig. 3, for each video frame $I_t$, we construct its DW-Mesh $M_t = \{V, F, T, O\}$, where $V$ represents vertices, $F$ denotes faces, $T$ represents mesh textures, and $O$ indicates whether faces are occluded. The construction process involves the following steps:

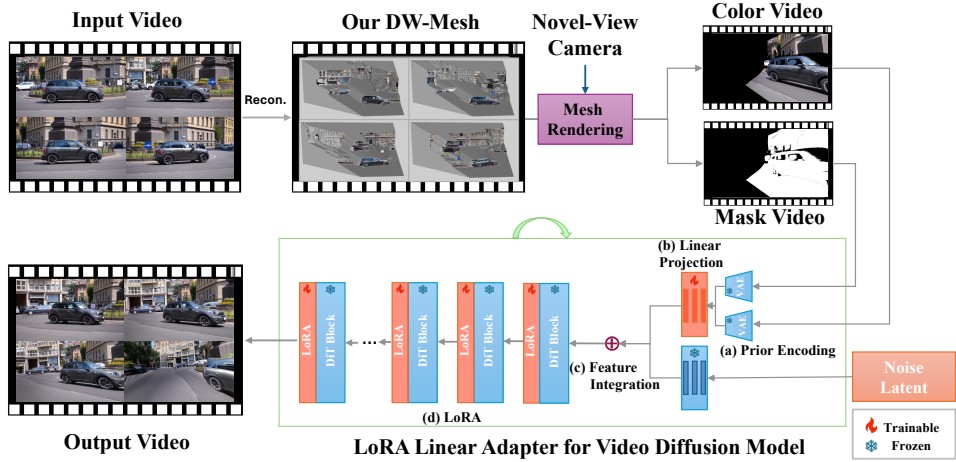

Figure 2: Overview of the EX-NVS framework. Our approach transforms monocular videos into extreme novel view videos through three key components: (1) Depth Watertight Mesh construction, which explicitly models both visible and occluded regions; (2) Color and mask videos are simulated or rendered for training or inference; and (3) a lightweight LoRA-based video diffusion adapter that ensures geometric consistency and temporal coherence in the synthesized videos.

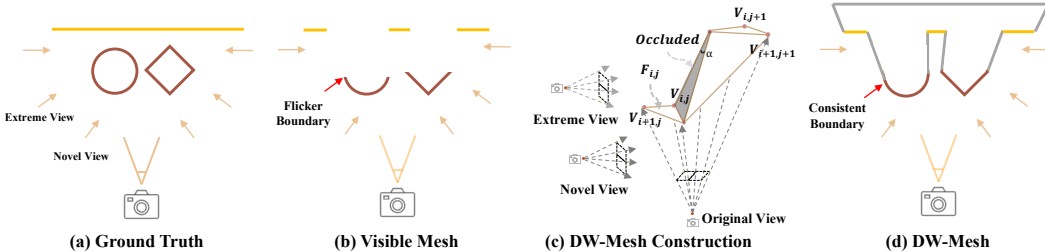

Figure 3: Illustration of DW-Mesh construction. (a) Ground Truth: The original scene with complete geometry. (b) Visible Mesh: 3D reconstructed visible mesh representation showing only the visible regions, causing flicker boundaries during rendering. (c) DW-Mesh Construction: We model both visible and invisible surfaces using watertight mesh from depth maps. (d) DW-Mesh: The watertight mesh representation ensures boundary consistency in extreme viewpoints.

**Vertex and Face Construction.** We compute per-frame depth maps $D_t$ using a pre-trained video depth estimation model (Hu et al., 2025). Each pixel $(i, j)$ with depth value $D_{i,j}$ is unprojected into 3D space to form a vertex $V_{i,j} = o + D_{i,j} \cdot r_{i,j}$, where $o$ is the canonical camera origin and $r_{i,j}$ is the ray direction. Triangular faces are constructed by connecting adjacent vertices in 2×2 grids. Boundary padding ensures watertight properties by setting $D_{\max}$ for frame-border pixels (details in appendix).

**Occlusion and Texture.** Rather than directly assigning pixel color as texture, we add an occlusion attribute. For each face, we perform geometric validation through minimum face angle analysis and depth discontinuity detection. Faces with minimum angle less than $\delta_{\text{angle}}$ or large depth discontinuities $\Delta D > \delta_{\text{depth}}$ are marked as occluded ($O_{i,j} = 1$), with texture set to $[0, 0, 0]$. Otherwise, faces use pixel color $C_{i,j}$ as texture. This produces a watertight mesh $M_t = \{V, F, T, O\}$ capturing both visible and occluded regions.

### 3.1.2 DW-MESH RENDERING

The DW-Mesh is rendered from the target camera trajectory $\{P_t\}_{t=1}^T$ to produce color and mask videos $V_T, V_O$ from mesh texture $T$ and occlusion attribute $O$. These outputs serve as geometric

priors, conditioning the video diffusion module to synthesize novel-view frames with improved visual consistency and geometric accuracy, even under challenging camera movements.

## 3.2 MASK GENERATION FOR TRAINING

Training video diffusion models for extreme viewpoint synthesis is challenging due to the scarcity of multi-view dynamic video datasets. To address this, we introduce a simulated masking strategy that creates effective training pairs from monocular videos without relying on paired multi-view data. This strategy includes two key components: Rendering Mask Generation and Tracking Mask Generation, as illustrated in Fig. 4.

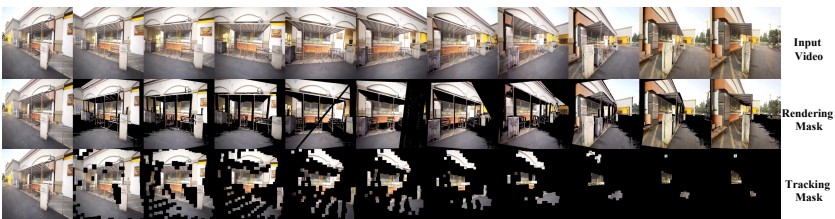

Figure 4: Illustration of our mask generation methods. Top Row: Input Monocular Video; Middle Row: Rendering Mask Generation uses DW-Mesh to simulate occlusions that would occur in novel viewpoints; Bottom Row: Tracking Mask Generation preserves temporal consistency by tracking points across frames and marking consistent occlusion patterns.

**Rendering Mask Generation.** This component leverages the DW-Mesh to generate realistic occlusion masks for novel viewpoints. We: 1) Construct DW-Mesh from input video and identify boundary mesh faces; 2) Render DW-Mesh under comprehensive rotation trajectories spanning -90° to 90° to produce binary visibility masks; 3) Apply morphological dilation to suppress noise while preserving structural integrity. This creates physically grounded occlusion masks for training.

**Tracking Mask Generation.** To ensure temporal consistency, we track feature points across frames using CoTracker3 (Karaev et al., 2024). We establish 10-50 points per frame and track their trajectories. When tracked points become occluded, we generate corresponding mask regions, creating temporally coherent occlusion transitions.

## 3.3 A LIGHTWEIGHT ADAPTER FOR VIDEO DIFFUSION

To synthesize realistic appearances for novel viewpoints, we build upon a pre-trained image-to-video diffusion model (et al., 2025b) and introduce a lightweight adapter. Our architecture integrates geometric priors through: (a) Encoding rendered color/mask videos via frozen VAE; (b) Linear projection to align with diffusion dimensions; (c) Feature integration by adding projected features to noise latents; (d) LoRA-based adaptation for efficient training with frozen backbone.

**LoRA Linear Adapter.** Our contribution extends standard LoRA by introducing a linear adapter that efficiently aggregates multiple control inputs (color and mask videos). While LoRA handles single tasks effectively, it cannot process multiple control signals simultaneously. Our novel linear adapter enables efficient aggregation through simple injection (x = x + latent), overcoming ControlNet's limitations like struggle in handling multiple control videos. The training objective follows standard diffusion denoising: $\mathcal{L} = \mathbb{E}_{\epsilon,t}[\omega(t)||\epsilon_\theta(z_t, I_1, V_T, V_O, t; \theta) - \epsilon||_2^2]$.

**Temporal Consistency Analysis.** Our approach ensures robust temporal and geometric coherence through three mechanisms: (1) *Temporally Consistent Depth Estimation*: We employ DepthCrafter (Hu et al., 2025), a state-of-the-art video depth estimator that maintains temporal smoothness across frames. While minor depth inconsistencies may occasionally occur, this limitation is shared across all depth-based methods and continues to improve as monocular depth estimation advances. (2) *Geometric Prior Filtering*: Our DW-Mesh representation explicitly filters unreliable geometric regions through occlusion-aware masking, providing stable surface priors that minimize temporal flickering

Table 1: Quantitative comparison of FID and FVD across viewpoint ranges.

| Method | FID↓ | | | FVD↓ | | |
|--------|------|------|------|------|------|------|
| | Small (0°→30°) | Large (0°→60°) | Extreme (0°→90°) | Small (0°→30°) | Large (0°→60°) | Extreme (0°→90°) |
| TrajectoryAttention (Xiao et al., 2025) | 59.86 | 62.69 | 62.49 | 623.54 | 754.80 | 912.14 |
| ReCamMaster (Bai et al., 2025) | 50.88 | 56.49 | 64.68 | 659.29 | 714.62 | 943.45 |
| TrajectoryCrafter (YU et al., 2025) | 48.72 | 55.24 | 65.33 | 633.25 | 725.44 | 893.80 |
| EX-NVS (Ours) | 44.19 | 50.30 | 55.42 | 571.18 | 685.39 | 823.61 |

Table 2: Quantitative comparison between methods in VBench metrics for the Full range (-90°→90°).

| Method | Aesthetic Quality ↑ | Imaging Quality ↑ | Temporal Flickering ↑ | Motion Smoothness ↑ | Subject Consistency ↑ | Background Consistency ↑ | Dynamic Degree ↑ |
|--------|---------|---------|---------|---------|---------|---------|---------|
| TrajectoryAttention (Xiao et al., 2025) | 0.389 | 0.567 | 0.895 | 0.931 | 0.834 | 0.846 | 0.923 |
| ReCamMaster (Bai et al., 2025) | 0.434 | 0.582 | 0.909 | 0.938 | 0.831 | 0.849 | 0.941 |
| TrajectoryCrafter (YU et al., 2025) | 0.447 | 0.607 | 0.902 | 0.928 | 0.838 | 0.856 | 0.936 |
| EX-NVS (Ours) | 0.450 | 0.631 | 0.914 | 0.934 | 0.846 | 0.872 | 0.948 |

in boundaries and maintain physical consistency across extreme viewpoint transitions. (3) *Diffusion-Based Temporal Modeling*: The video diffusion backbone employs self-attention mechanisms that process frame sequences holistically, enabling effective propagation of appearance and geometric information across time to ensure coherent motion dynamics and structural consistency.

## 4 EXPERIMENTS

### 4.1 EXPERIMENTAL SETTINGS

**Datasets.** For training, we utilize OpenVID (Nan et al., 2024), a large-scale monocular video dataset with over 1 million high-quality videos spanning diverse scenes and motion patterns. For comprehensive evaluation, we construct a challenging testing dataset of 150 carefully selected in-the-wild videos, comprising 100 static scenes and 50 dynamic scenes with varying complexity levels. We evaluate across four progressive angular ranges to assess performance scalability: Small (0°→30°), Large (0°→60°), Extreme (0°→90°), and Full (-90°~90°), enabling thorough analysis of method robustness under increasingly challenging viewpoint changes.

**Metrics.** We employ a comprehensive evaluation protocol using multiple complementary metrics: FID (Heusel et al., 2017a) for assessing visual quality and realism, FVD (Heusel et al., 2017b) for evaluating temporal coherence and video dynamics, VBench (Huang et al., 2024) for comprehensive perceptual quality assessment across multiple dimensions, and structured user studies for human perceptual evaluation. For 3D consistency validation, we use novel view synthesis metrics with 3D Gaussian Splatting reconstruction. All methods use identical camera trajectories and depth inputs for rigorous and fair comparison.

**Baselines.** We compare against state-of-the-art camera-controllable video synthesis methods from both major paradigms: geometry-based approaches including TrajectoryCrafter (YU et al., 2025) and TrajectoryAttention (Xiao et al., 2025), and camera-parameter-based conditioning methods such as ReCamMaster (Bai et al., 2025). These baselines represent the current leading approaches in novel view video synthesis.

**Implementation Details.** Training uses Wan2.1 (et al., 2025b) (14B parameters) as frozen backbone with 140M trainable adapter parameters. We use LoRA rank 16, AdamW optimizer with lr $3 \times 10^{-5}$, training on 32 A100 GPUs for 24 hours. Videos are $512 \times 512$ resolution with 49 frames. Inference uses 25 denoising steps, taking 4 minutes per video.

### 4.2 QUANTITATIVE COMPARISON

**Video Quality.** The results in Table 1 demonstrate that our EX-NVS consistently outperforms all baselines across different metrics and viewpoint ranges. For FID scores, our method achieves 44.19, 50.30, and 55.42 for small, large, and extreme viewpoint ranges respectively, showing significant improvements over the second-best method (TrajectoryCrafter with 48.72, 55.24 for small and

large ranges, and TrajectoryAttention with 62.49 for extreme angles). Similarly, ours achieves the lowest FVD scores (571.18, 685.39, and 823.61) across all viewpoint ranges, demonstrating superior temporal coherence compared to the baselines. Notably, as the viewpoint angles become more extreme, the performance gap widens, highlighting our method's robustness in handling challenging camera movements.

Table 2 shows our method achieves the highest scores on most VBench (Huang et al., 2024) metrics, including aesthetic quality (0.450), imaging quality (0.631), and temporal consistency (0.914). Our scores for subject consistency (0.846) and background consistency (0.872) demonstrate the geometric stability of our DW-Mesh representation. Our method maintains consistent quality across extreme camera movements, confirming the effectiveness of our DW-Mesh approach. We also evaluated both static and dynamic scenes separately (results in supplementary), showing consistent outperformance across diverse scene types. Table 3 further breaks down performance on static vs dynamic scenes under extreme viewpoints (0°→90°). Our method consistently outperforms all baselines in both scenarios, highlighting the versatility of our DW-Mesh representation for handling diverse content types.

**Scene type performance.** To demonstrate the robustness of our approach across different scene types, we evaluate our method separately on static and dynamic scenes. Table 3 shows that EX-NVS consistently outperforms all baselines in both scenarios, highlighting the versatility of our DW-Mesh representation for handling diverse content types.

**3D Consistency in 6DoF using NVS.** Our method supports full 6-degree-of-freedom camera motion. Evaluation on arbitrary translation and rotation combinations shows superior performance with larger margins than pure rotational movements, confirming DW-Mesh's robustness across complex trajectories (details in appendix). To comprehensively assess the geometric consistency of our method, we evaluate 3D consistency via Novel View Synthesis metric by reconstructing 3D scenes with 3D Gaussian Splatting (Kerbl et al., 2023) from generated videos and computing PSNR between generated frames and rendered novel views. As shown in Table 4, our method achieves the highest PSNR of 28.09, significantly outperforming all baselines and demonstrating superior 3D consistency in camera-driven video generation.

Table 3: Scene type performance (0°→90°).

| Method | Static | | Dynamic | |
|---|---|---|---|---|
| | FID↓ | FVD↓ | FID↓ | FVD↓ |
| TrajectoryAttention | 66.24 | 974.32 | 66.78 | 967.85 |
| ReCamMaster | 68.92 | 1007.21 | 69.08 | 1001.45 |
| TrajectoryCrafter | 69.65 | 953.94 | 69.77 | 958.12 |
| EX-NVS (Ours) | 59.14 | 879.72 | 58.96 | 881.34 |

Table 4: 3D Consistency in 6DoF using NVS.

| Method | PSNR ↑ |
|---|---|
| TrajectoryAttention (Xiao et al., 2025) | 18.45 |
| ReCamMaster (Bai et al., 2025) | 19.65 |
| TrajectoryCrafter (YU et al., 2025) | 24.17 |
| EX-NVS (Ours) | 28.09 |

## 4.3 QUALITATIVE COMPARISON

We present comprehensive qualitative comparisons in Fig. 5 across diverse challenging scenarios. Existing geometry-based approaches (TrajectoryCrafter and TrajectoryAttention) show fundamental limitations with extreme viewpoints, producing severe ghosting artifacts, geometric distortions, and inconsistent object boundaries due to their inability to properly model hidden surfaces and handle occlusion transitions. These methods often fail to maintain object shape integrity under large viewpoint changes, resulting in warped or disconnected structures. ReCamMaster exhibits inconsistent object boundaries, temporal flickering, and struggles with extreme camera trajectories, often producing unrealistic viewpoint deviations outside its training distribution.

In stark contrast, our EX-NVS method produces physically consistent videos with superior occlusion handling, maintaining object shapes, spatial relationships, and temporal coherence even under the most challenging extreme camera movements. Our DW-Mesh representation ensures smooth occlusion transitions and prevents the geometric artifacts that plague existing methods, resulting in visually convincing and temporally stable extreme viewpoint videos. Additional qualitative results spanning diverse scene types are provided in the supplementary material.

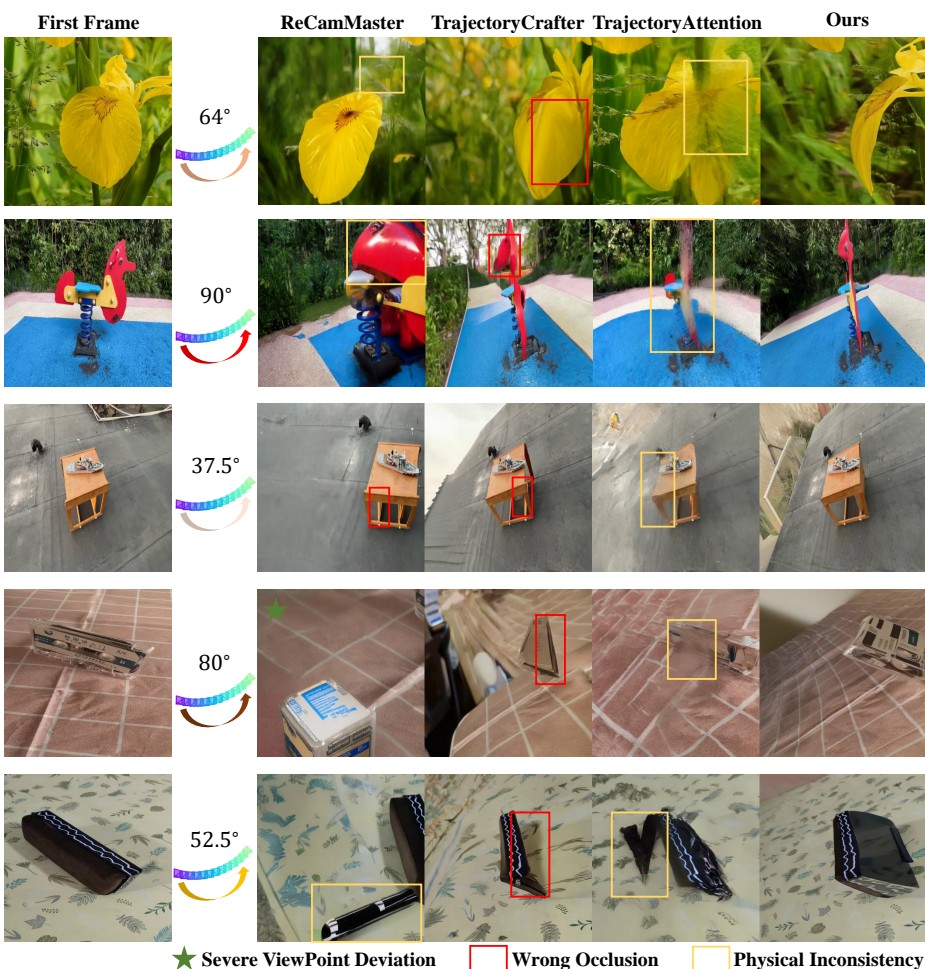

**First Frame**   **ReCamMaster**   **TrajectoryCrafter**   **TrajectoryAttention**   **Ours**

★ **Severe ViewPoint Deviation**   □ **Wrong Occlusion**   □ **Physical Inconsistency**

Figure 5: Qualitative comparison under extreme viewpoints. Our approach produces physically consistent videos with effective occlusion handling and temporal coherence. In contrast, baselines exhibit artifacts such as **physical inconsistency** (shape/scale distortion), **wrong occlusion** (leaking/ghosting near boundaries), and **severe viewpoint deviation** in scenes outside their training distribution.

Table 5: Ablation study on the components of EX-NVS in **Extreme Viewpoint (0°→90°)**.

| Variant | FID↓ | FVD↓ |
|---|---|---|
| Full Method (DW-Mesh + LoRA Rank 16 + Wan2.1) | **55.42** | **823.61** |
| w/o DW-Mesh | 74.31 (worse 34.085%) | 1103.21 (worse 33.948%) |
| w/ Random Masks | 69.36 (worse 25.153%) | 993.64 (worse 20.644%) |
| w/o Rendering Masks | 63.35 (worse 14.309%) | 972.93 (worse 18.130%) |
| w/o Tracking Masks | 60.24 (worse 8.697%) | 924.47 (worse 12.246%) |
| w/ LoRA Rank 64 | 53.68 (better 3.140%) | 802.47 (better 2.567%) |

## 4.4 USER STUDY

We conducted a comprehensive user study with 50 participants evaluating 12 video sets across diverse scenes and camera movements. Participants selected the method with best physical consistency and extreme viewpoint synthesis quality. As shown in Fig. 6, our method received 70.70% preference compared to TrajectoryCrafter (14.96%), ReCamMaster (9.50%), and TrajectoryAttention (4.84%), demonstrating superior handling of complex occlusions and temporal transitions.

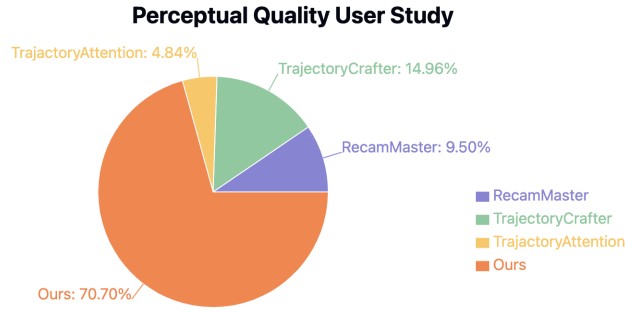

Figure 6: User study results comparing our EX-NVS method against baselines. Participants evaluated videos based on quality of physical consistency and extreme viewpoint, with our approach receiving significantly higher preference ratings.

## 4.5 ABLATION STUDY

**Model contribution.** Table 5 shows the impact of each component. Removing the DW-Mesh (W/o DW-Mesh) leads to the largest performance drop (FID: +34.1%, FVD: +33.9%), underscoring its critical importance for extreme viewpoint synthesis. Using random masks significantly degrades performance (FID: +25.2%, FVD: +20.6%), demonstrating the value of our structured geometric guidance over arbitrary masking strategies. Both rendering and tracking masks are crucial for optimal performance, with rendering masks providing stronger geometric constraints and tracking masks ensuring temporal consistency. Switching from Wan2.1 to smaller backbones reduces performance, but our method still outperforms baselines. Increasing LoRA rank from 16 to 64 yields minimal improvements, confirming our lightweight design is already effective.

**Backbone Model Fairness Analysis.** To address fairness concerns, we tested our method with smaller backbones. Table 6 shows that even with Wan2.1-1.3B (1.3B parameters), our method (FID: 62.12, FVD: 878.95) outperforms TrajectoryCrafter with CogVideoX-5B (FID: 65.33, FVD: 893.80), confirming improvements stem from DW-Mesh representation rather than model scale.

Table 6: Backbone model fairness comparison.

| Method | FID↓ | FVD↓ |
|---|---|---|
| **TrajectoryCrafter w/ CogVideoX-5B** (YU et al., 2025) | 65.33 | 893.80 |
| **EX-NVS w/ Wan2.1-1.3B** | 62.12 | 878.95 |
| **EX-NVS w/ CogVideoX-5B** | 59.76 | 867.25 |
| **EX-NVS w/ Wan2.1-14B** | 55.42 | 823.61 |

## 5 CONCLUSION

We introduced EX-NVS, a framework for generating high-quality videos from monocular input under extreme viewpoints. Our key innovation, the Depth Watertight Mesh (DW-Mesh), ensures geometric consistency by explicitly modeling visible and occluded regions. Our simulated masking strategy eliminates the need for multi-view training data, while the lightweight LoRA adapter efficiently integrates geometric priors into video diffusion models. Extensive experiments demonstrate consistent outperformance of state-of-the-art methods, with particularly significant improvements at extreme camera angles. User studies confirm superior perceptual quality and physical consistency.

**Limitations:** Our framework relies on depth estimation quality, may struggle with fine structures, and requires significant computation for high-resolution generation. Future work will focus on involving dynamic scene reconstruction and improving model efficiency.

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

# A MORE IMPLEMENTATION DETAILS

## A.1 DW-MESH CONSTRUCTION DETAILS

For vertex and face construction, triangular faces are formed using:

$$F_{i,j,1} = \{(i,j),(i+1,j),(i,j+1)\}, \tag{1}$$
$$F_{i,j,2} = \{(i+1,j),(i+1,j+1),(i,j+1)\}. \tag{2}$$

Two additional boundary faces $\{(0,0),(0,W),(H,0)\}$ and $\{(H,0),(H,W),(0,W)\}$ ensure complete watertight mesh construction.

For occlusion detection, the criteria are:

$$O_{i,j} = \begin{cases} 1, & \text{if } \text{Min}(\angle(F_{i,j})) < \delta_{\text{angle}} \text{ or } \Delta D > \delta_{\text{depth}}, \\ 0, & \text{otherwise.} \end{cases} \tag{3}$$

The texture assignment follows:

$$T_{i,j} = \begin{cases} [0,0,0], & \text{if } O_{i,j} = 1, \\ C_{i,j}, & \text{otherwise,} \end{cases} \tag{4}$$

## A.2 DETAILED TRAINING PROCESS

The video diffusion model parameters $\theta$ are defined as: $\epsilon$: ground-truth noise, $z_t$: noisy latents, $\omega(t)$: training weight at timestep $t$, $I_1$: first frame, $\epsilon_\theta$: denoising model.

The adapter uses LoRA rank 16 across attention layers (q, k, v, o) and feed-forward blocks (ffn.0, ffn.2). Training uses AdamW optimizer with detailed hyperparameters in supplementary material.

## A.3 NETWORK STRUCTURE

The EX-NVS Adapter consists of four main modules: Prior Encoding, Linear Projection, Feature Integration, and LoRA. Below, we provide detailed descriptions of each module:

**Prior Encoding** leverages a frozen Video VAE encoder from Wan Text-to-Vodeo model (et al., 2025b) to extract compact latent representations from both the input color video and the corresponding mask video. Specifically, given input sequences of shape $\mathbb{R}^{49 \times 512 \times 512}$, the VAE encodes each into latent tensors of shape $\mathbb{R}^{7 \times 64 \times 64}$, where 49 is the number of frames and $512 \times 512$ is the spatial resolution. This encoding preserves essential spatiotemporal information while significantly reducing dimensionality, enabling efficient downstream processing. The encoded latents from the color and mask videos are then concatenated along the channel dimension to form a unified geometric prior, which is subsequently fed into the linear projection module for further feature transformation.

**Linear Projection** is implemented as a sequence of $1 \times 1 \times 1$ Conv3d layers followed by a final Conv3d layer with kernel size $(1, 2, 2)$ and stride $(1, 2, 2)$. The concatenated latent features from the prior encoding stage are first projected to a higher-dimensional hidden space using the $1 \times 1 \times 1$ convolutions with SiLU activations. The final Conv3d layer then downsamples the spatial dimensions to produce patch embeddings that match the expected input shape of the diffusion model. This design ensures efficient channel mixing and spatial alignment between the geometric priors and the video diffusion backbone.

The following code snippet illustrates the implementation of both prior encoding and linear projection layer within the EX-NVS adapter:

```
import torch
import torch.nn as nn

class PriorEncoding(nn.Module):
    """
    A VAE model for encoding camera information and video features.
    """

```

```python
9    def __init__(
10        self,
11        in_channels: int = 16,
12        hidden_channels: int = 1024,
13        out_channels: int = 5120,
14    ) -> None:
15        super().__init__()
16
17        self.latent_encoder = torch.nn.Sequential(
18            torch.nn.Conv3d(in_channels * 2, hidden_channels,
         kernel_size=1, stride=1, padding=0),
19            torch.nn.SiLU(),
20            torch.nn.Conv3d(hidden_channels, hidden_channels,
         kernel_size=1, stride=1, padding=0),
21            torch.nn.SiLU(),
22            torch.nn.Conv3d(hidden_channels, hidden_channels,
         kernel_size=1, stride=1, padding=0)
23        )
24        self.latent_patch_embedding = torch.nn.Conv3d(hidden_channels,
         out_channels, kernel_size=(1, 2, 2), stride=(1, 2, 2))
25        nn.init.zeros_(self.latent_patch_embedding.weight)
26        nn.init.zeros_(self.latent_patch_embedding.bias)
27
28    def _set_gradient_checkpointing(self, module, value=False):
29        if isinstance(module, nn.Module):
30            module.gradient_checkpointing = value
31
32    def forward(self, video, mask, vae) -> torch.Tensor:
33        with torch.no_grad():
34            video = vae.encode(video, device=video.device)
35            mask = vae.encode(mask * 2 - 1, device=mask.device)
36        latent = torch.cat([video, mask], dim=1)
37        latent = self.latent_encoder(latent)
38        latent = self.latent_patch_embedding(latent)
39        return latent
40
41 def prepare_camera_embeds(
42     prior_encoding,
43     vae,
44     video,
45     mask=None,
46 ) -> torch.Tensor:
47     prior_latent = prior_encoding(video, mask, vae)
48     return prior_latent
```

Listing 1: EX-NVS Adapter: Prior Encoding and Linear Projection.

**Feature Integration** fuses the projected geometric priors with the noise latent features used in the diffusion process. The integration is performed by element-wise addition, allowing the model to condition the generation process on both the appearance and occlusion information encoded in the priors. This design enables the adapter to inject geometric consistency and mask-aware guidance into the video synthesis pipeline.

```python
1 import torch
2 import torch.nn as nn
3 % === EX-NVS: Prior Encoding ===
4 x = self.patch_embedding(noise_latent)
5 prior_latent = prior_encoding(video, mask, vae)
6
7 % === Start: EX-NVS Adapter: Feature Integration ===
8 x = self.patch_embedding(noise_latent)
9 x = x + prior_latent
10 % === End: EX-NVS Adapter: Feature Integration ===
11
12 % === EX-NVS: Diffusion Transformer ===
```

```
13  x = self.transformer(x, context, time_embedding)
```

Listing 2: EX-NVS Adapter: Feature Integration.

**LoRA (Low-Rank Adaptation)** is employed to enable efficient fine-tuning of the adapter with minimal trainable parameters. In our implementation, LoRA layers are applied to the following modules: q, k, v, o, ffn.0, and ffn.2 within each attention block of the video diffusion backbone. The LoRA module introduces low-rank updates to these linear projection weights, allowing the adapter to adapt to new tasks or domains without updating the full set of backbone parameters. This approach significantly reduces memory and computational consumption, making the EX-NVS Adapter lightweight and scalable for large-scale video generation tasks.

Together, these modules enable the EX-NVS Adapter to effectively incorporate geometric priors and mask information into the video diffusion process, resulting in high-quality, physically consistent, and temporally coherent video synthesis under extreme viewpoints.

### A.4 LoRA INTEGRATION IN VIDEO DIFFUSION MODELS

We employ Low-Rank Adaptation (LoRA) to efficiently fine-tune our video diffusion backbone. The following Python function demonstrates how LoRA modules are injected into a model, targeting specific layers such as attention projections and feed-forward blocks. This approach enables parameter-efficient adaptation by updating only a small subset of weights.

```python
1   def add_lora_to_model(self, model, lora_rank=16, lora_alpha=16,
        lora_target_modules="q,k,v,o,ffn.0,ffn.2", init_lora_weights="
        kaiming", pretrained_path=None, state_dict_converter=None):
2       # Add LoRA to UNet
3       self.lora_alpha = lora_alpha
4       if init_lora_weights == "kaiming":
5           init_lora_weights = True
6
7       lora_config = LoraConfig(
8           r=lora_rank,
9           lora_alpha=lora_alpha,
10          init_lora_weights=init_lora_weights,
11          target_modules=lora_target_modules.split(","),
12      )
13      model = inject_adapter_in_model(lora_config, model)
14      for param in model.parameters():
15          # Upcast LoRA parameters into fp32
16          if param.requires_grad:
17              param.data = param.to(torch.float32)
18
19      # Lora pretrained lora weights
20      if pretrained_path is not None:
21          state_dict = load_state_dict(pretrained_path)
22          if state_dict_converter is not None:
23              state_dict = state_dict_converter(state_dict)
24          missing_keys, unexpected_keys = model.load_state_dict(
        state_dict, strict=False)
25          all_keys = [i for i, _ in model.named_parameters()]
26          num_updated_keys = len(all_keys) - len(missing_keys)
27          num_unexpected_keys = len(unexpected_keys)
28          print(f"LORA: {num_updated_keys} parameters are loaded from {
        pretrained_path}. {num_unexpected_keys} parameters are unexpected."
        )
```

Listing 3: EX-NVS Adapter: Feature Integration module.

This function configures and injects LoRA modules into the specified target layers, optionally loading pretrained LoRA weights. It ensures all trainable parameters are in float16 for numerical stability. This design allows for scalable and memory-efficient adaptation of large video diffusion models.

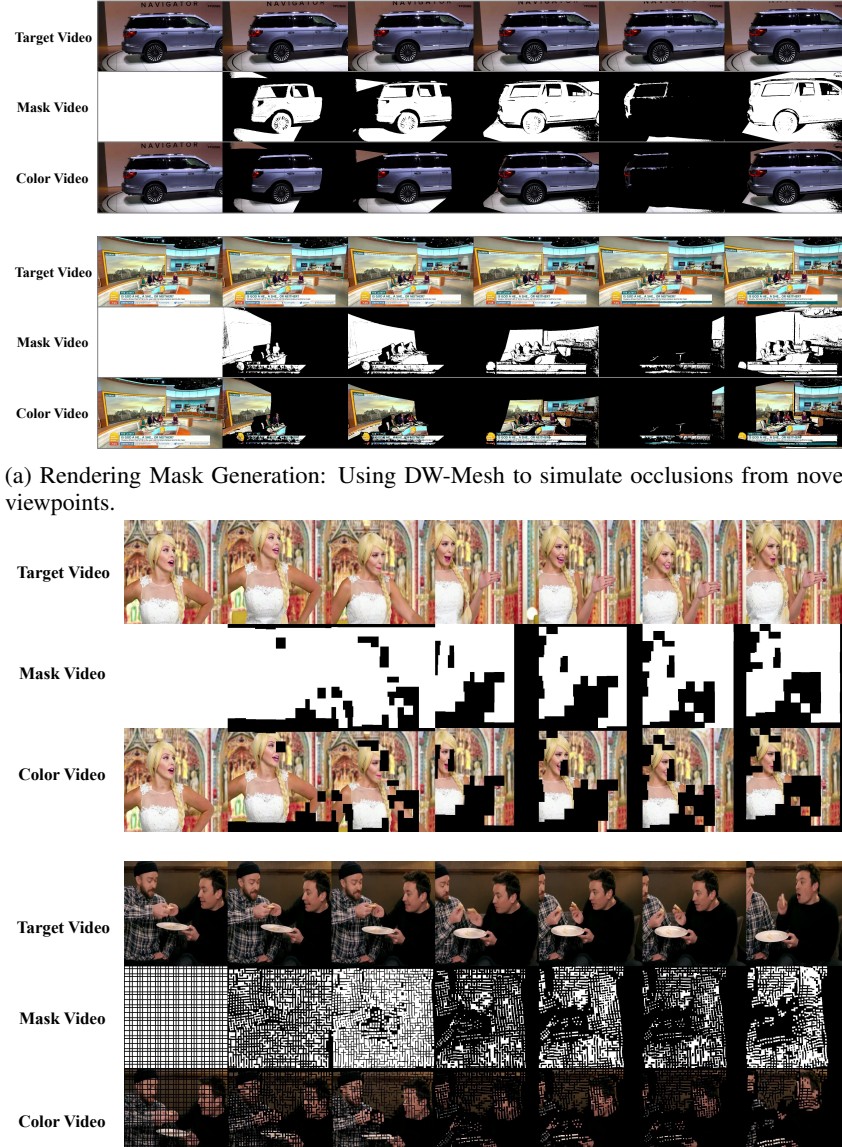

(a) Rendering Mask Generation: Using DW-Mesh to simulate occlusions from novel viewpoints.

(b) Tracking Mask Generation: Preserving temporal consistency through point tracking across frames.

Figure 7: Detailed visualization of our mask generation methods. (a) Rendering masks are created by simulating novel viewpoint occlusions using the DW-Mesh representation. (b) Tracking masks ensure temporally consistent occlusion patterns by tracking points across consecutive frames.

### A.5   DETAILS ABOUT MASK GENERATION

Fig. 7 illustrates more examples about our rendering and tracking mask approaches.

Rendering mask generation relies on uniform sampling of diverse viewpoint angles across the full -90° to 90° range, ensuring comprehensive coverage of potential camera positions during inference. This technique leverages the DW-Mesh representation to simulate realistic occlusions that would occur when viewing the scene from novel perspectives. To ensure the generation of realistic occlusion masks, we enforce adjacent faces $\{(i, j), (i+1, j), (i, j+1)\}$ and $\{(i+1, j), (i, j+1), (i+1, j+1)\}$ must be either simultaneously occluded or unoccluded. Subsequently, We apply morphological dilation

operation with the kernel size of $5 \times 5$ on the binary mask. This process effectively removes isolated noise pixels while preserving the structural integrity of major occlusion regions, ensuring smooth and continuous occlusion boundaries.

The tracking mask approach establishes a grid of 10-50 points per frame, with grid size randomly selected for each training instance to ensure model learning from varied point distributions. We maintain balance between spatial coverage and computational efficiency by adjusting density based on scene complexity. An off-the-shelf tracker (Karaev et al., 2024) follows points across consecutive frames, preserving consistent visibility patterns to simulate temporal occlusion effects. The principle of tracking mask generation is illustrated in Fig. 8.

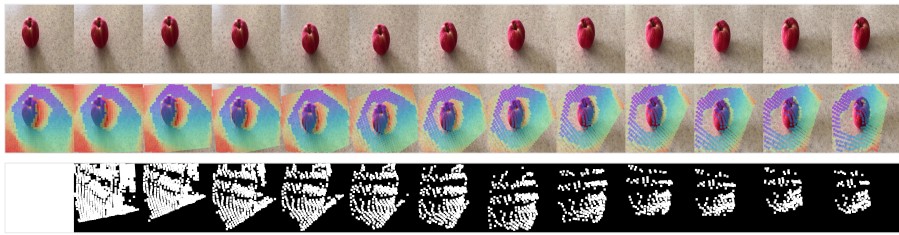

Figure 8: Principle of tracking mask generation. Points are tracked across frames to create consistent occlusion patterns, ensuring temporal coherence. Different colors represent corresponding tracked points between frames, helping maintain consistent visibility relationships during motion.

Additional video augmentation techniques enhance training diversity. Our smooth cropping procedure operates in both horizontal and vertical directions, using crop window sizes of 85-95% of the original frame. Rather than static crops, we generate smooth trajectories following Bezier curves with controlled acceleration and deceleration. This approach introduces viewpoint variations without requiring explicit 3D understanding, improving the model's ability to generalize to diverse camera movements.

## A.6 THE EFFECT OF DW-MESH

To evaluate the impact of the DW-Mesh representation, we conduct ablation studies comparing it with visible mesh-based methods (w/o DW-Mesh). As illustrated in Figure 9, our experiments demonstrate that DW-Mesh significantly improves occlusion handling and view synthesis quality, particularly in scenes with complex geometry and dynamic occlusions. Even our DW-Mesh occludes the background, it still generates high-quality results.

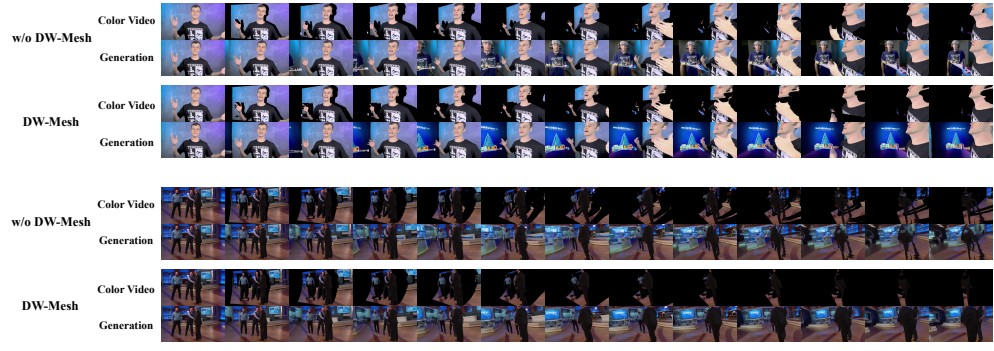

Figure 9: Comparison of DW-Mesh vs. w/o DW-Mesh under extreme viewpoints.

## A.7 FAILURE CASES

Despite EX-NVS's effectiveness for extreme viewpoint synthesis, several challenging scenarios can lead to suboptimal results:

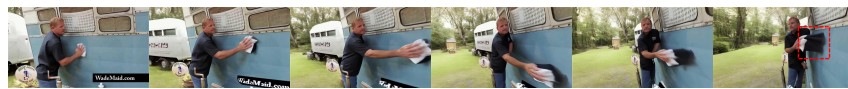

(a) Failure due to inaccurate depth estimation: incorrect geometry leads to distorted occlusion boundaries.

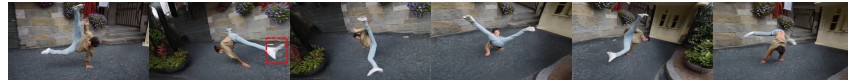

(b) Failure on fine/thin structures: mesh oversmoothing or missing thin objects causes loss of detail or floating artifacts.

Figure 10: Representative failure cases of EX-NVS. (a) Depth estimation errors causing visible distortions in novel views; (b) Fine structure handling limitations where thin objects are lost or misrepresented.

**Depth Estimation Limitations.** Our framework relies heavily on monocular depth estimation quality. When depth maps contain errors due to challenging scenes (reflective surfaces, complex lighting, rapid motion), the resulting DW-Mesh may exhibit geometric inaccuracy. As shown in Fig. 10a, these inaccuracies can propagate to synthesized views, causing visible distortions or incorrect occlusion boundaries.

**Fine Geometric Detail Preservation.** The watertight mesh construction process may struggle with very thin structures or fine details. Features like wires, fences, or small protruding elements might be oversmoothed or entirely missing in the reconstructed geometry. Fig. 10b demonstrates how this limitation can result in loss of detail or floating artifacts in rendered outputs.

### A.8 FUTURE WORK: DW-MESH REFINEMENT PROCESS

While EX-NVS demonstrates significant improvements in extreme novel view synthesis, DW-Mesh is still in its early stages and can benefit from further refinement. A promising future direction is developing an iterative refinement stage that can be applied after initial video synthesis. This refinement process would address potential temporal inconsistencies and improve background surface reconstruction quality through multi-pass optimization. The refinement would re-estimate depth maps from generated video frames, leveraging the temporal smoothing and geometric consistency enforced during video generation to produce more accurate depth estimates than the initial monocular predictions.

### A.9 USER STUDY SETTINGS

We conducted a comprehensive user study to evaluate the perceptual quality of our method compared to baseline approaches. The study involved 50 participants evaluating 12 randomly selected video sequences from our test dataset. Each video sequence contained results from our EX-NVS method and all three baseline approaches (ReCamMaster, TrajectoryCrafter, and TrajectoryAttention).

Participants were asked to select which method produced the most visually compelling results based on two key criteria: physical consistency (maintaining object integrity without unrealistic deformations) and extreme viewpoint quality (demonstrating significant camera movement with a strong sense of 3D space). As shown in Fig.11, the study interface presented videos in randomized order (labeled as Methods A-D) to avoid position bias.

To ensure reliable results, we included attention check questions and allowed participants to replay videos multiple times before making selections. The results, as presented in Fig. 6, showed a strong preference for our method, with 70.70% of participants selecting EX-NVS as producing the most physically consistent and convincing extreme viewpoint videos.

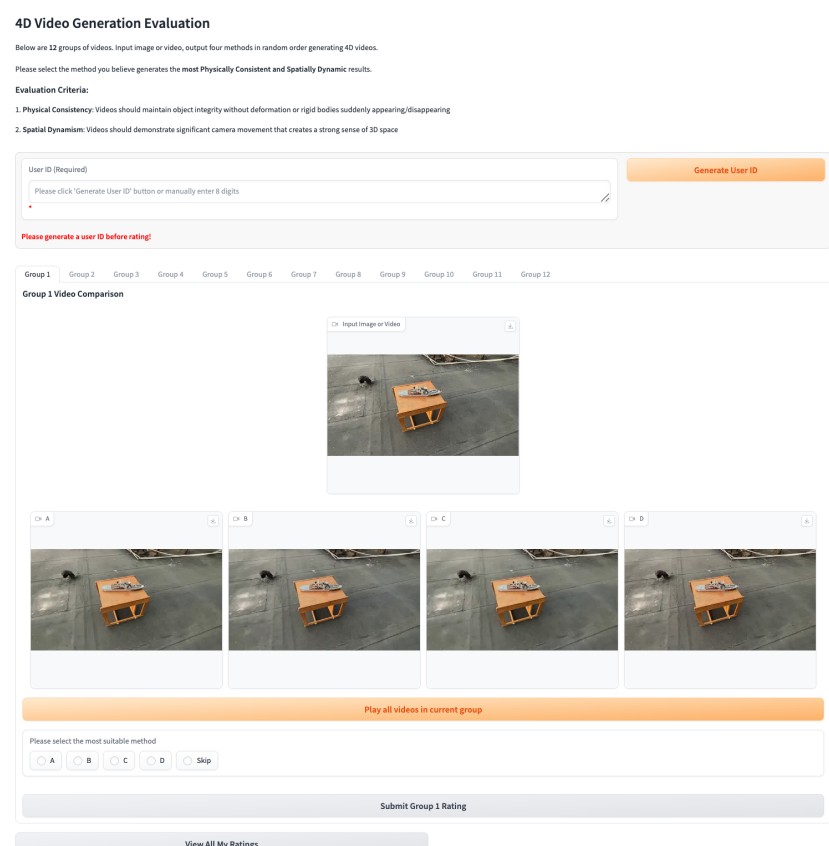

Figure 11: User study interface. Participants were presented with four methods (labeled A-D in randomized order) and asked to select the one that produced the most physically consistent and convincing extreme viewpoint videos. The interface allowed for multiple viewings before selection to ensure informed comparisons.

### A.10 MORE VISUALIZATION

We provide additional visual comparisons between our EX-NVS method and state-of-the-art approaches. Fig. 12, Fig. 13. Fig. 14, Fig. 15, Fig. 16 and Fig. 17 show results across diverse scenes and challenging camera trajectories.

### A.11 ADDITIONAL ABLATION STUDY DETAILS

In addition to the main ablation results presented in Table 5, we provide additional implementation details and analysis:

**Masking Strategy Analysis.** Both rendering and tracking masks are crucial—removing rendering masks increases FID by 14.3% and FVD by 18.1%, while removing tracking masks raises FID by 8.7% and FVD by 12.2%. The combination of both strategies provides optimal performance.

**LoRA Rank Analysis.** Increasing the LoRA rank from 16 to 64 yields only slight improvements (3.1% on FID, 2.6% on FVD), indicating our lightweight adapter with rank 16 is already effective and efficient, providing a good balance between performance and computational efficiency.

**Implementation Parameter Details.** We set the depth threshold $\delta_{\text{depth}} = 0.013 \left( \max(D_t) - \min(D_t) \right)$ per frame, and $D_{\max} = 100$ for boundary padding. Nvdiffrast (Laine et al., 2020) is

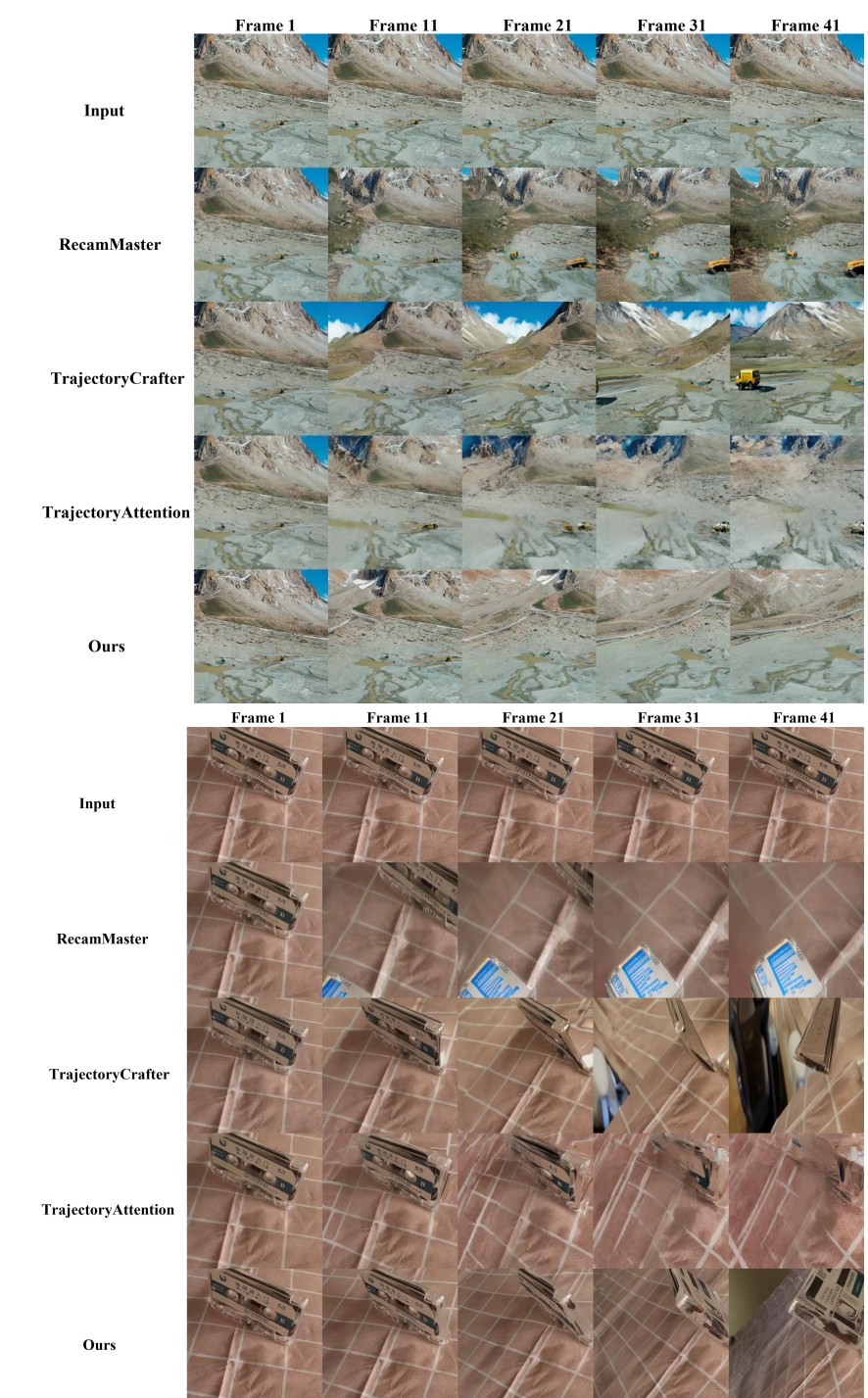

Figure 12: Comparison of EX-NVS with state-of-the-art methods.

adopted as the renderer for both training and validation. Input videos are resized to $512 \times 512$ with 49 frames per sequence, and we use 25 denoising steps during inference.

## A.12 DETAILED ALGORITHM DESCRIPTIONS

Due to space constraints in the main paper, we provide the detailed algorithmic descriptions here.

---

**Algorithm 1** DW-Mesh Construction (per frame)

---

**Require:** RGB frame $I_t$, depth map $D_t$, camera origin $o$, pixel rays $\{r_{i,j}\}$, thresholds $\delta_{\text{angle}}$, $\delta_{\text{depth}}$, boundary depth $D_{\text{max}}$
1: Pad frame-border pixels in $D_t$ with $D_{\text{max}}$
2: Unproject each pixel: $V_{i,j} \leftarrow o + D_{i,j} \cdot r_{i,j}$
3: Form triangular faces on $2 \times 2$ grids; add two large boundary triangles
4: For each face $F_{i,j}$, compute min angle and depth discontinuity $\Delta D$
5: Set occlusion: $O_{i,j} \leftarrow \mathbb{1}[\min \angle(F_{i,j}) < \delta_{\text{angle}} \ \vee \ \Delta D > \delta_{\text{depth}}]$
6: Set texture: $T_{i,j} \leftarrow \begin{cases} [0,0,0] & O_{i,j} = 1 \\ C_{i,j} & \text{otherwise} \end{cases}$
7: **return** $M_t = \{V, F, T, O\}$

---

**Algorithm 2** Simulated Mask Generation for Training

---

**Require:** Video $\{I_s\}_{s=1}^{S}$, target trajectory $\{P_t\}_{t=1}^{T}$, DW-Mesh renderer, dilation kernel $\mathcal{K}$
1: Build DW-Mesh $\{M_s\}$ from $\{I_s\}$ (Alg. 1)
2: Render visibility masks $\{m_t\}$ along $\{P_t\}$ from $\{M_s\}$
3: Denoise masks via morphological dilation: $m_t \leftarrow m_t * \mathcal{K}$
4: Track points across frames and zero-out local rectangles when occluded
5: Compose final mask video $V_O$ and masked color video $V_T$
6: **return** $(V_T, V_O)$

---

To address concerns about our method's performance on different types of camera movements, we conducted comprehensive evaluations comparing translational versus rotational viewpoint changes. While our primary focus is on demonstrating extreme-view performance, our method is capable of generating free navigation across all translational and rotational movements. The demo video included in the supplementary material showcases arbitrary camera motion in 6DOF.

We performed additional evaluations comparing our method with state-of-the-art approaches under arbitrary combinations of camera translation and rotation. We constrain the trajectory of camera within the range:

**Camera Trajectory Constraints:**

- **Camera position**: $x_{cam} \in [-r/1.3, r/1.3], y_{cam} \in [-r/2, r/4], z_{cam} \in [-0.5, r - 0.1]$

- **Look-at point**: $x_{lookat} \in [-r/7, r/7], y_{lookat} \in [-r/4, r/4], z_{lookat} \in [r - 0.1, r + 0.1]$

where $r$ is the minimum depth value of the first frame. This random strategy prevents drastic view shifts while providing diverse 6DOF camera motions, ensuring robust evaluation across different motion types.

## A.13 DEMO VIDEO

We provide a demo video showcasing the capabilities of our EX-NVS framework in the supplementary file. Using scenes synthesized by SOTA video generation models such as Veo3, Sora, and Kling, we generate highly physically consistent novel views under extreme and complex camera trajectories. The results highlight the effectiveness of our approach in generating high-quality, temporally consistent videos under extreme viewpoints.

## A.14 USE OF LARGE LANGUAGE MODELS

We confirm that large language models (LLMs) were used only for minor editorial and coding assistance (grammar, phrasing, and clarity). They were not involved in formulating ideas, designing the method, running or analyzing experiments, drafting technical sections, or drawing conclusions. All scientific contributions, experiments, analyses, and interpretations are solely by the authors.

## B    REPRODUCIBILITY STATEMENT

To ensure the reproducibility of our results, we have provided comprehensive implementation details and supporting materials. The core source code was provided in the supplementary material. We also detailed our training procedures, hyperparameters, and evaluation protocols.

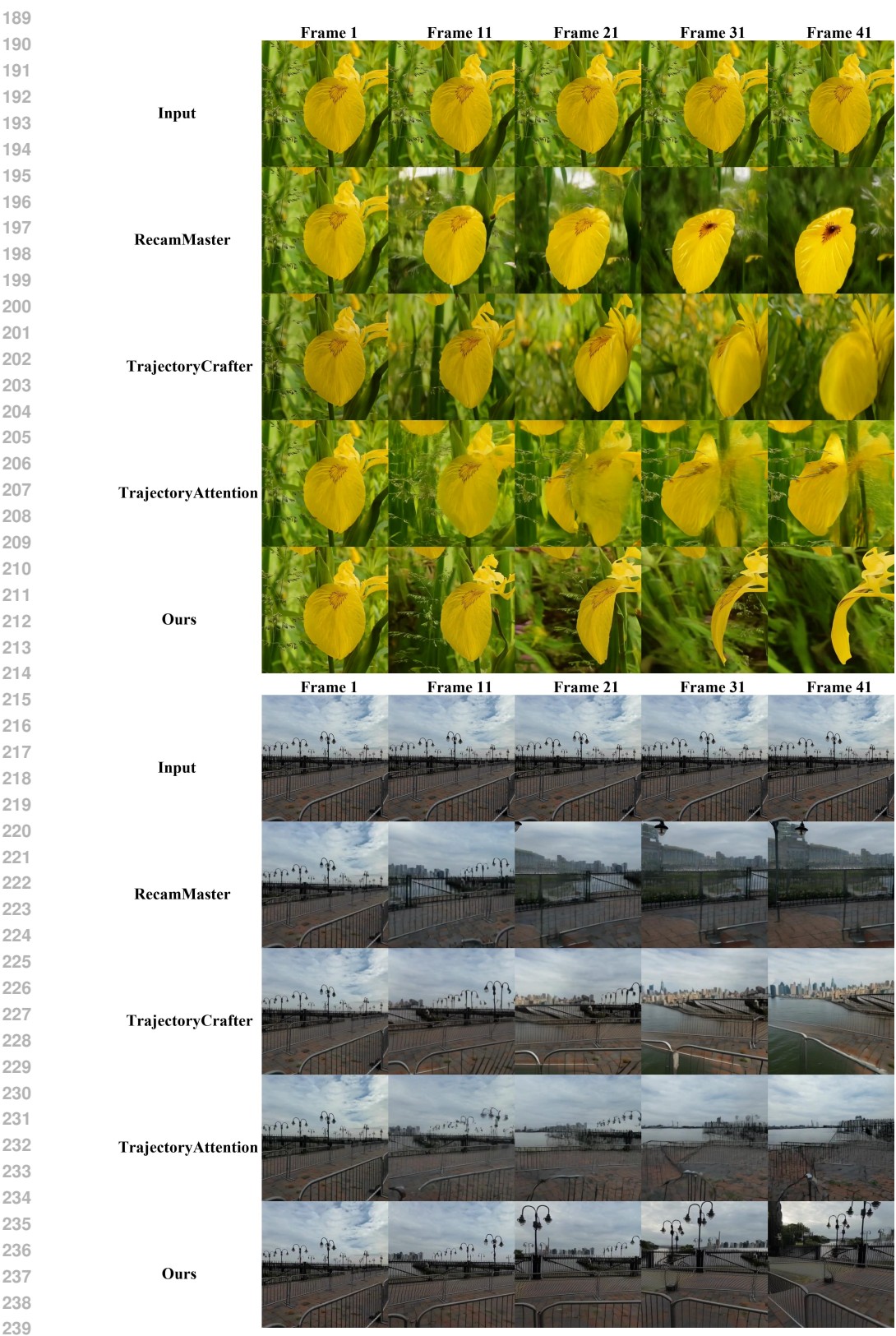

Figure 13: Comparison of EX-NVS with state-of-the-art methods.

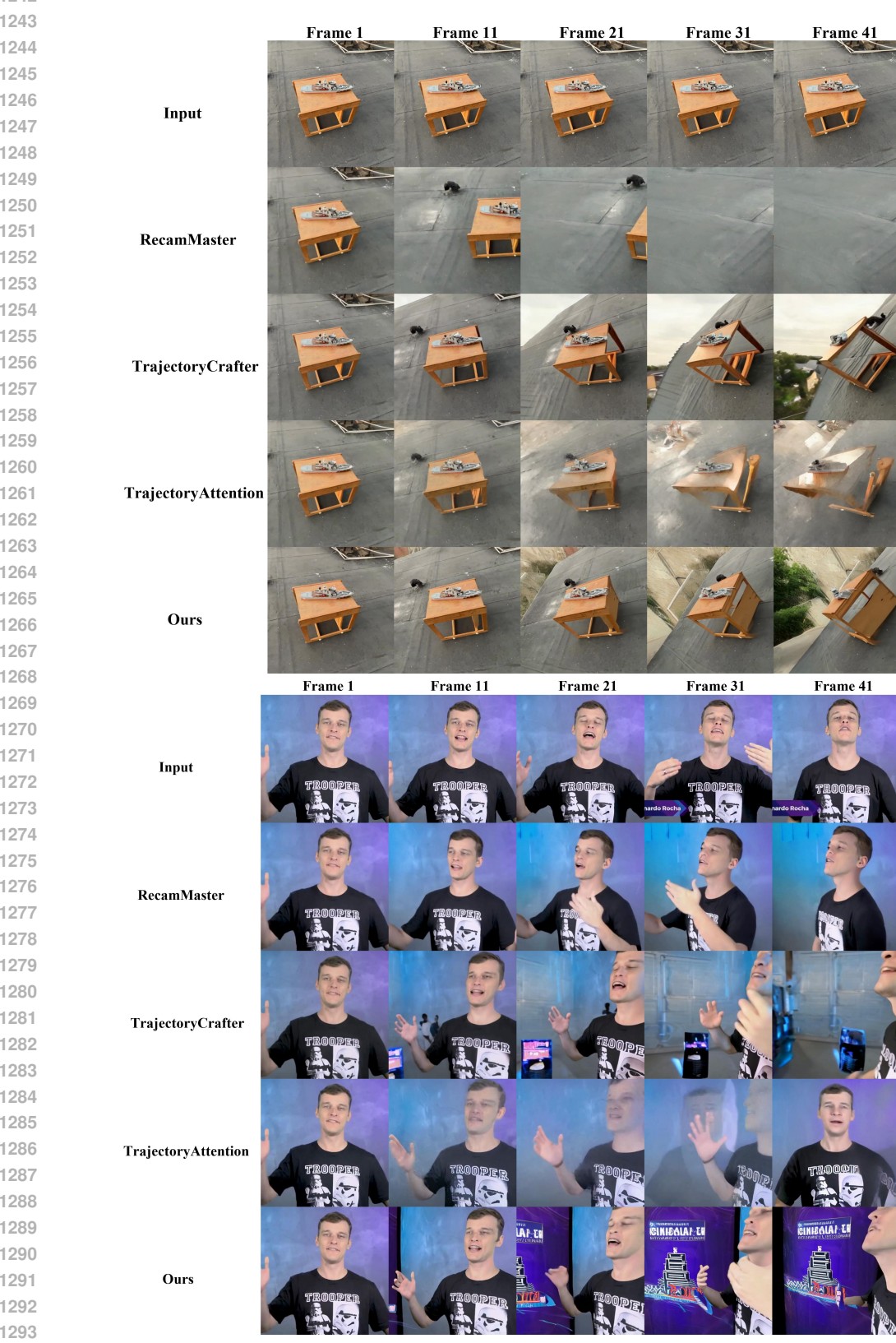

Figure 14: Comparison of EX-NVS with state-of-the-art methods.

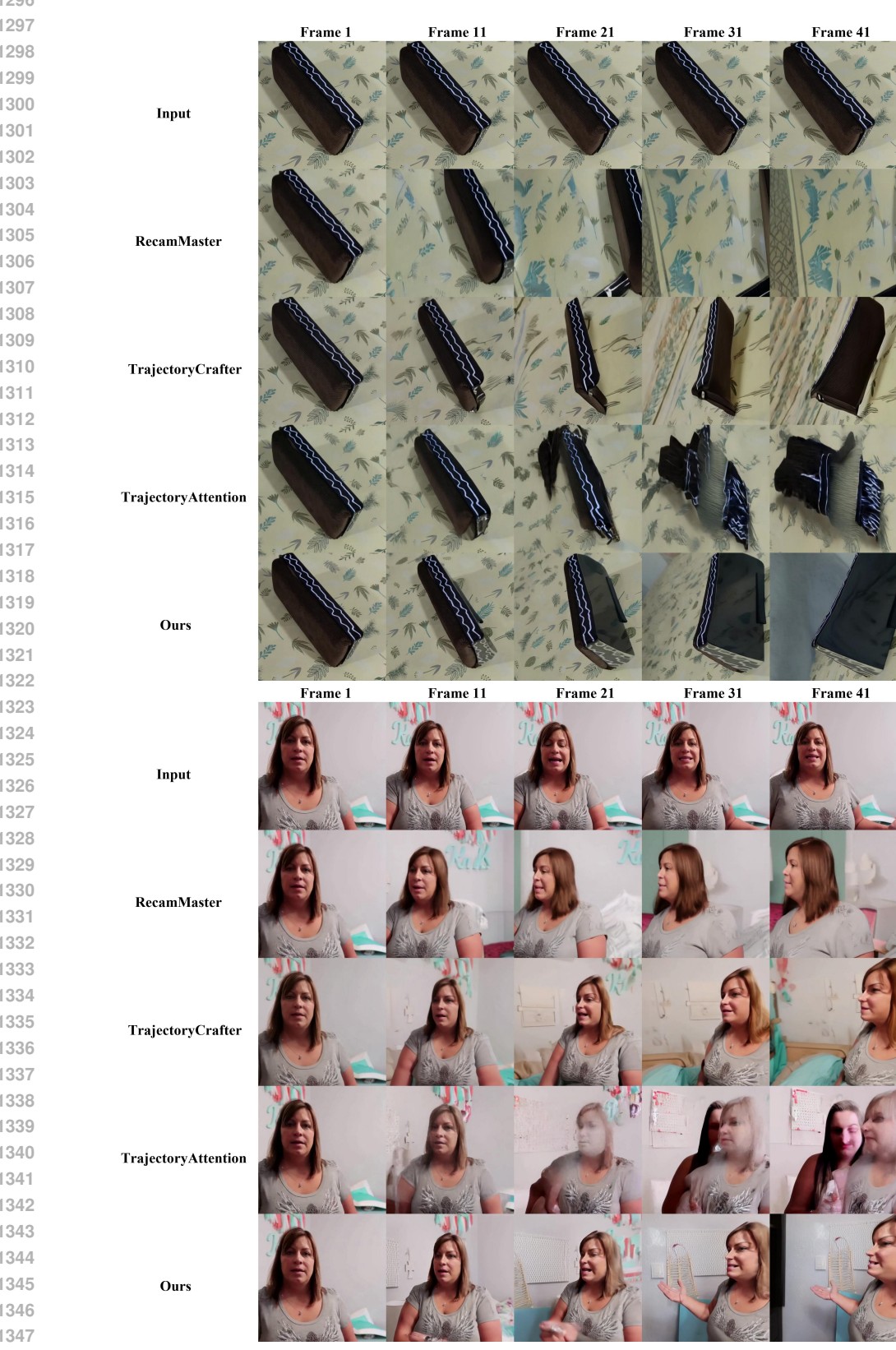

Figure 15: Comparison of EX-NVS with state-of-the-art methods.

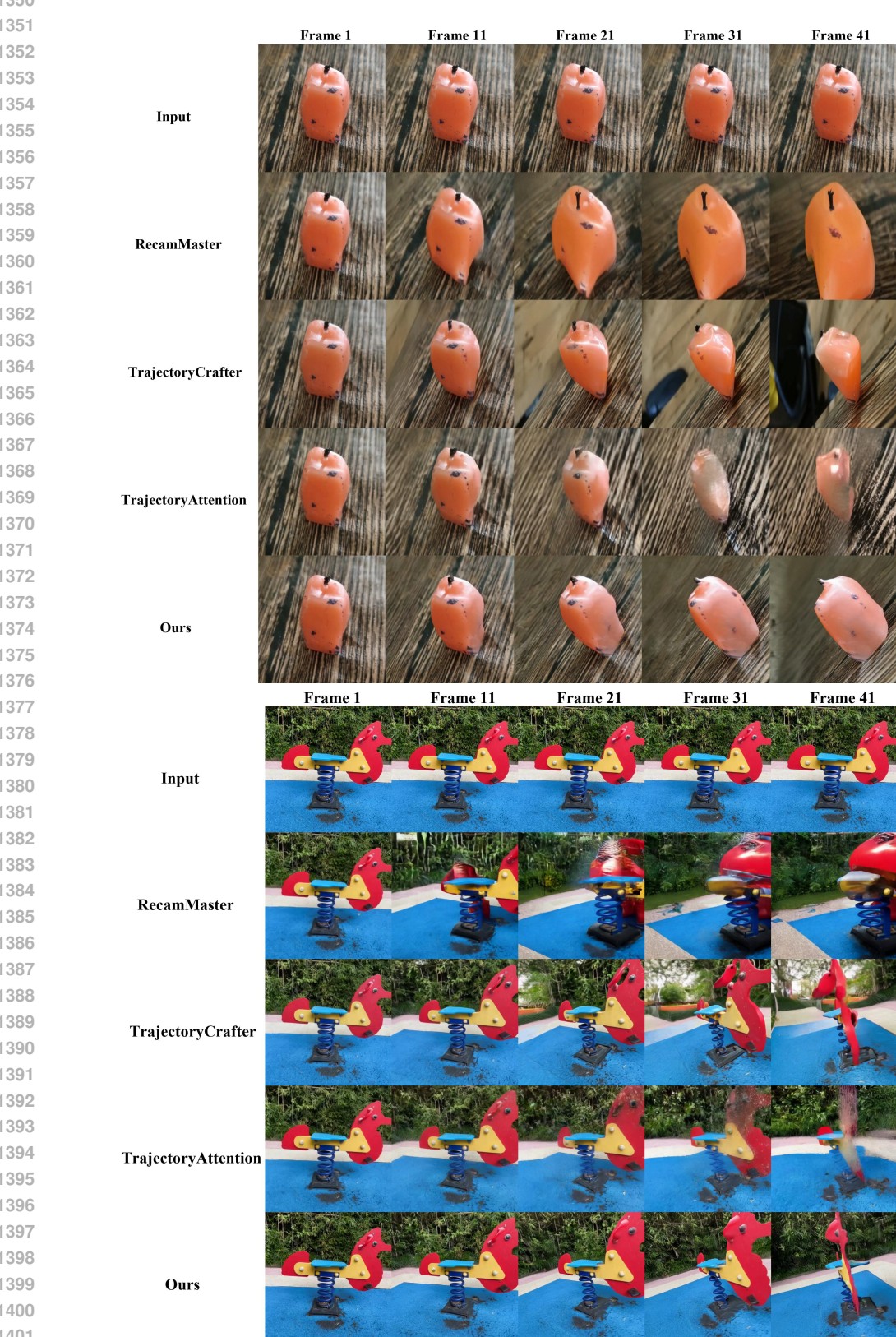

Figure 16: Comparison of EX-NVS with state-of-the-art methods.

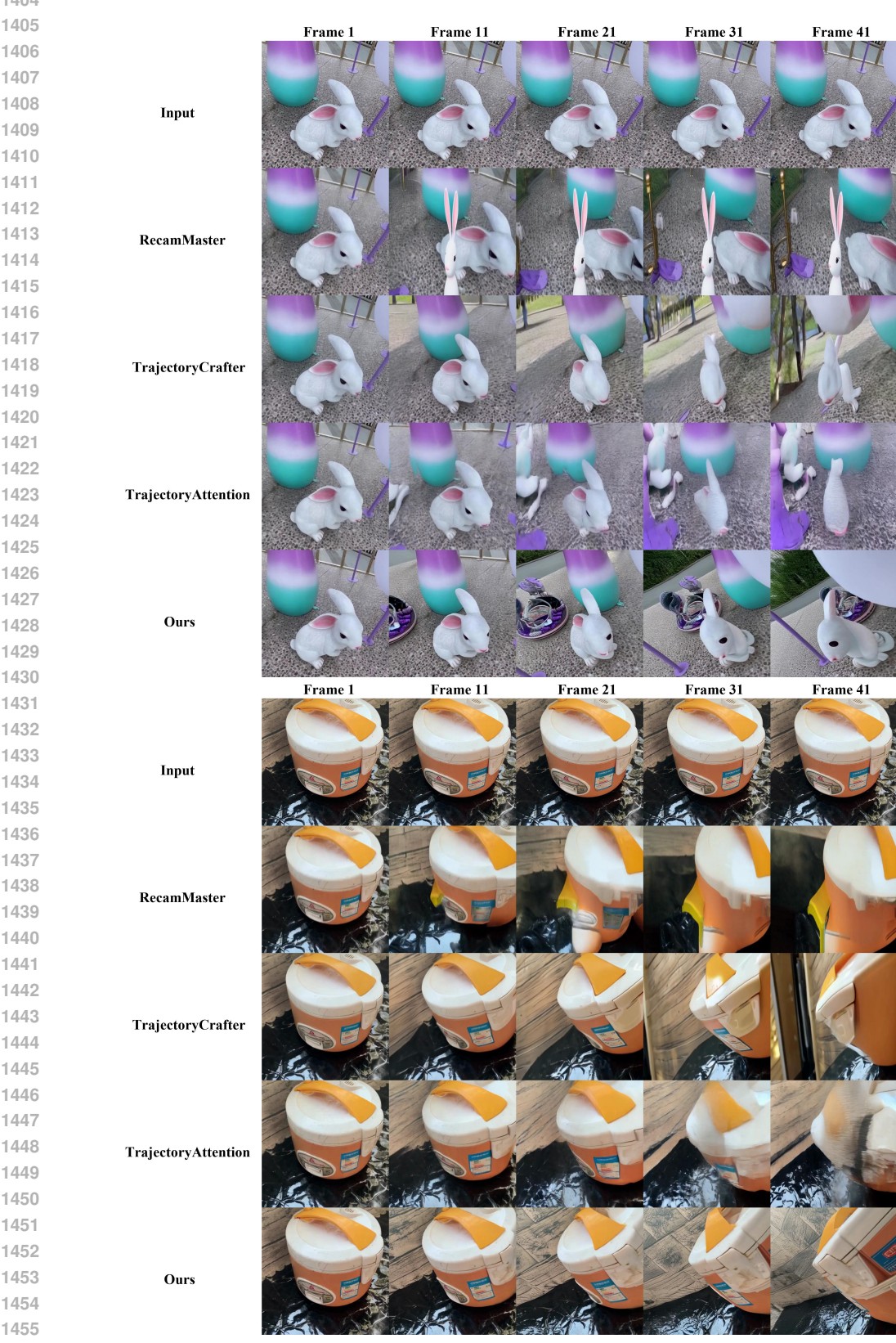

Figure 17: Comparison of EX-NVS with state-of-the-art methods.

