# OpenReview forum: "EX-NVS: EXtreme Novel View Synthesis via Depth Watertight Mesh"
_ICLR.cc/2026/Conference — ICLR 2026 Conference Withdrawn Submission_

### Official Review · Reviewer_SMKz · 2025-10-30

**Soundness:** 3
**Presentation:** 2
**Contribution:** 2
**Rating:** 4
**Confidence:** 4

**Summary:**

This paper introduces EX-NVS, a framework designed to address the challenging problem of extreme novel view synthesis (NVS) from a single monocular video. The core technical contribution is the proposed Depth Watertight Mesh (DW-Mesh) representation, which explicitly models both visible and occluded regions to provide a robust geometric prior for rendering, thereby ensuring geometric consistency under out-of-distribution viewpoints. The authors further incorporate a linear LoRA adapter to enhance the diffusion model component. Experimental results demonstrate the method’s ability to generate novel views of dynamic scenes.

**Strengths:**

1. **Addressing a Challenging Problem**: Generating photorealistic and geometrically consistent novel views from a monocular dynamic video, especially for "extreme" viewpoints, is a crucial and highly difficult problem in 3D computer vision. The paper’s objective is ambitious and addresses a key limitation of existing NVS methods.

2. **Novel Geometric Representation**: The concept of using a Depth Watertight Mesh (DW-Mesh) is an interesting approach to explicitly model scene geometry, particularly for handling difficult boundary conditions and occlusions.

3. **Use of Diffusion Models**: The integration of a diffusion model, combined with a LoRA adapter, to improve the realism and quality of the generated images is a sound strategy given the current state-of-the-art in generative video diffusion models.

**Weaknesses:**

1. **Clarity and Motivation of Technical Contributions**: The benefits and motivations behind the two core contributions—DW-Mesh and the linear LoRA adapter—are not sufficiently clear or compelling. The paper should better articulate why these components are essential and what specific challenges they address that existing methods cannot. For example, what advantages does DW-Mesh offer over point clouds, which are more efficient to construct and provide higher geometric precision? Similarly, why is a linear LoRA adapter used instead of a standard LoRA or direct fine-tuning? The claim that "LoRA cannot process multiple control signals simultaneously" should be further explained and justified.

2. **Unsatisfactory Visual Results**: The visual quality shown in the results video suggests a relatively low performance ceiling, as noticeable distortions are present in several cases. This raises concerns about either the accuracy of the proposed DW-Mesh or the effectiveness of the linear LoRA component. The authors should investigate and clarify the root causes of these artifacts.

3. **Abstract**: The opening of the abstract appears truncated or incomplete; the phrase “these challenges” lacks clear context, which hinders the reader’s immediate understanding of the paper’s main contributions.

4. **Novelty Concerns**: The mask creation process appears structurally similar to the reprojection process of TrajectoryCrafter, raising concerns about novelty. This aspect should be clarified and better differentiated to highlight the paper’s unique contributions.

5. **Weak and Incomplete Evaluation**: The experimental validation is a major weakness, particularly concerning the lack of standard and robust metrics for novel view synthesis (NVS).
- The paper lacks the crucial standard image quality metrics for NVS, namely SSIM and LPIPS. The visual assessment of NVS performance heavily relies on these, in addition to PSNR.

- The PSNR results presented in Table 4 are deficient in experimental details. The authors must explicitly state which datasets were used for this evaluation. Whether the evaluation was conducted using a rigorous benchmark like DyCheck[1], which is essential for properly evaluating monocular dynamic NVS.

- The reliance on fitting a 3D Gaussian Splatting (3DGS) or similar structure to evaluate 3D consistency is insufficient for evaluating novel view synthesis. PSNR, SSIM, and LPIPS can be computed directly on the rendered novel views against ground truth images for a proper NVS evaluation, which seems to be lacking or is confusingly presented.

[1] Hang Gao, Ruilong Li, Shubham Tulsiani, Bryan Russell,
and Angjoo Kanazawa. Monocular dynamic view synthesis: A reality check. In NeurIPS, 2022.

**Questions:**

1. **Technical Justification**: Can the authors provide a more detailed and quantitative ablation study that isolates the precise performance gain contributed by (a) the Depth Watertight Mesh and (b) the linear LoRA adapter? Please quantify the improvement in terms of geometric consistency (e.g., fidelity of the recovered mesh) and rendering quality (PSNR/SSIM/LPIPS).

2. **Mask Creation Novelty**: Please explicitly describe how the mask creation process differs from established methods like TrajectoryCrafter, detailing the unique features or adaptations that justify it as a novel component of EX-NVS.

---

### Official Review · Reviewer_wY2E · 2025-10-30

**Soundness:** 3
**Presentation:** 3
**Contribution:** 3
**Rating:** 4
**Confidence:** 3

**Summary:**

The paper proposes EX-NVS, a method for generating high-quality videos from monocular inputs under extreme viewpoints. It introduces the Depth Watertight Mesh (DW-Mesh) to model both visible and occluded regions, ensuring complete and watertight geometry. To overcome the lack of multi-view data, a simulated masking strategy is used for supervision. Experiments show that EX-NVS effectively handles occlusions and maintains geometric consistency, producing realistic and coherent results.

**Strengths:**

1. Generating camera-controllable videos for dynamic scenes under extreme viewpoints remains an unsolved problem. The proposed DW-Mesh is well-motivated, addressing geometry-based approaches’ limitations in rendering such challenging views.

2. Qualitative results of extreme-viewpoint rendering show that the method effectively handles unobserved regions and achieves improvements over baseline methods.

3. Video demonstrations are provided to visualize the synthesized results.

**Weaknesses:**

1. The level of novelty is limited, as the overall pipeline closely resembles TrajectoryCrafter, with the main difference being the replacement of its point cloud representation by the proposed DW-Mesh. Moreover, the simulated masking strategy for addressing data scarcity is conceptually similar to that used in TrajectoryCrafter.

2. The construction of DW-Mesh depends on accurate depth and camera pose estimation, and thus inherits the fragility of geometry-based approaches when such estimations are unreliable.

3. The submission lacks side-by-side comparison videos, which would help better assess qualitative differences against baseline methods.

**Questions:**

1. The authors claim that this work represents a paradigm shift bridging camera-based and geometry-based methods. However, the proposed pipeline still fundamentally follows a geometry-based framework — reconstructing explicit 3D geometry (DW-Mesh) from input frames and rendering these representations from target viewpoints to guide generation. How this design conceptually connects to or integrates principles from camera-based approaches?

2. During DW-Mesh construction, the process requires camera parameters to unproject depth maps into 3D space. How are these camera poses obtained?

---

### Official Review · Reviewer_583o · 2025-11-01

**Soundness:** 3
**Presentation:** 3
**Contribution:** 3
**Rating:** 6
**Confidence:** 4

**Summary:**

This paper presents EX-NVS, a framework for generating novel view videos from monocular input under extreme camera viewpoints. The main contribution is a Depth Watertight Mesh (DW-Mesh) representation that explicitly models visible and occluded regions to maintain geometric consistency for viewpoint changes. The authors propose a simulated masking strategy to overcome the scarcity of multi-view training data. Finally, the method employs a LoRA diffusion adapter for video synthesis. Experiments demonstrate quantitative improvements over state-of-the-art methods.

**Strengths:**

- The watertight mesh representation addresses a very commonly present limitation of the existing methods. This approach provides a more complete geometric prior than others.
- The training strategy with rendering and tracking masking is very sound and seems to be a significant contribution, making it possibly useful for other methods as well.
- The evaluation of similar works uses several different evaluation protocols. The authors here make a good effort to establish a solid evaluation protocol for extreme novel view synthesis.
- The ablation in Table 5 successfully emphasises the weight of the specific contributions.
- The results from Tables 1 and 2 show a notable quantitative improvement.
- The qualitative examples included in the paper look convincing.

**Weaknesses:**

- Some results on the novel view synthesis dataset would be great, e.g. DyCheck, which is used in evaluation by TrajectoryCrafter.
- Similarly, while I like the evaluation setup, the authors should follow the setup of a prior work (e.g. ReCamMaster) and then propose their setup in addition for a fairer and direct comparison.
- It would be interesting to see the effect of depth estimation on the performance, e.g. through using GT and estimated depth (could be done on DyCheck dataset). Further, different depth estimators would make for a good ablation.

**Questions:**

- Were all the compared methods trained on the same data?
- Is the mesh representation struggling with reflections or with changing lighting conditions?
- I would like to see more details on the setup in paragraph '3D Consistency in 6DoF using NVS'.
- In the user study, how were the users selected?
- What is the breakdown of inference time between each component, and how does it scale with video length and resolution?

---

### Official Review · Reviewer_gy1j · 2025-11-01

**Soundness:** 2
**Presentation:** 3
**Contribution:** 2
**Rating:** 4
**Confidence:** 4

**Summary:**

The paper “EX-NVS: Extreme Novel View Synthesis via Depth Watertight Mesh” presents a framework for generating extreme-angle videos (−90° to 90°) from monocular input. The method introduces a Depth Watertight Mesh (DW-Mesh) representation that explicitly models both visible and occluded regions to maintain geometric consistency. To compensate for missing multi-view data, the authors simulate training supervision through rendering and tracking masks, and train a LoRA-based video-diffusion adapter that integrates the DW-Mesh priors for temporally coherent synthesis. Experiments show better FID/FVD and perceptual scores compared with camera- and geometry-based baselines such as TrajectoryCrafter, ReCamMaster, and TrajectoryAttention.

**Strengths:**

+ Interesting target problem. Extreme-viewpoint synthesis is difficult, and explicit occlusion reasoning is relevant to the field of 3D-aware video generation.

+ Clear presentation and quantitative results. The paper is well written and provides detailed ablations (DW-Mesh, mask variants, LoRA rank) and fairness checks across different backbones.

+ Comprehensive evaluation. Quantitative (FID, FVD, VBench) and user-study evaluations support the claim that geometric priors improve physical consistency.

**Weaknesses:**

- Limited conceptual novelty. The proposed “Depth Watertight Mesh’’ is essentially a dense depth-to-mesh conversion with simple boundary padding, which is conceptually close to existing mesh or depth-map fusion techniques used in multi-view reconstruction. The paper repackages standard depth-map meshing and occlusion labeling under new terminology rather than introducing a genuinely new geometric formulation.

- Simulated-mask strategy is incremental. The rendering- and tracking-mask idea mimics visibility-map and optical-flow consistency widely explored in view-synthesis and self-supervised video learning. It is more a data-augmentation trick than a principled solution to the absence of multi-view data.

- Heavy pipeline and dependency chain. The framework relies on a pre-trained depth estimator (DepthCrafter), external tracker (CoTracker3), LoRA fine-tuning on a huge video backbone (Wan 2.1), and additional mask generation. This multi-stage system is complex to apply, and its novelty mainly lies in combining existing components rather than proposing new algorithms.

- Ablation analysis lacks mechanistic insight. While ablations report performance drops when removing components, there is no deeper analysis of why DW-Mesh helps (e.g., no visualization of latent-space geometry, no causal explanation linking watertightness to temporal stability).

**Questions:**

Could a simpler learned depth-completion or NeRF-style implicit surface achieve similar occlusion handling without explicit meshing?

---

### Note · Authors · 2025-11-13

I have read and agree with the venue's withdrawal policy on behalf of myself and my co-authors.